# Optimization of Laccase/Mediator System (LMS) Stage Applied in Fractionation of *Eucalyptus globulus*

**DOI:** 10.3390/polym11040731

**Published:** 2019-04-22

**Authors:** Javier M. Loaiza, Ascensión Alfaro, Francisco López, María T. García, Juan C. García

**Affiliations:** Research Center in Technology of Products and Chemical Processes, PRO2TECS- Chemical Engineering Department, Campus “El Carmen”, University of Huelva, 21071 Huelva, Spain; javiermauricio.loiza@diq.uhu.es (J.M.L.); baldovin@uhu.es (F.L.); mtrinidad.garcia@diq.uhu.es (M.T.G.); juan.garcia@diq.uhu.es (J.C.G.)

**Keywords:** autohydrolysis, *Eucalyptus globulus*, laccase, hemicellulose, pulp, soda-AQ

## Abstract

In a biorefinery framework, a laccase/mediator system treatment following autohydrolysis was carried out for eucalyptus wood prior to soda-anthraquinone pulping. The enzymatic and autohydrolysis conditions, with a view to maximizing the extraction of hemicelluloses while preserving the integrity of glucan, were optimized. Secondly, pulping of solid phase from *Eucalyptus globulus* wood autohydrolysis and the enzymatic process was carried out and compared with a conventional soda-anthraquinone (AQ) pulping process. The prehydrolysis and enzymatic delignification of the raw material prior to the delignification with soda- Anthraquinone (AQ) results in paper sheets with a lower kappa number and brightness and strength properties close to conventional soda-AQ paper and a liquid fraction rich in hemicellulose compounds that can be used in additional ways. The advantage of this biorefinery scheme is that it requires a lower concentration of chemical reagents, and lower operating times and temperature in the alkaline delignification stage, which represents an economic and environmental improvement over the conventional process.

## 1. Introduction

Paper mills have had to comply with increasingly stringent environmental regulations by developing new, less polluting technologies, which have produced less waste and allowed for the more efficient use of material resources and energy over the last few decades. In parallel, the silvicultural sector has sought new methods to valorize fractions that were formerly disposed of as waste or that interfered with other industrial processes. These two trends have converged in biorefinery processes, a new approach to the obtainment of a wide range of commercial products in addition to cellulose pulp and paper [1,2,3,4,5,6]. Biorefining processes typically involve chemical and/or biochemical fractionation treatments [7].

Some biorefining processes use enzymes or fungi for pulp and paper production. Biopulping pretreatments and enzyme-based biobleaching are known to provide substantial advantages. For example, biochemical mediators are neither toxic nor chemically hazardous. Additionally, they can be used under the usual temperature conditions of paper mills [8,9,10], reduce energy use in subsequent steps such as pulp beating, and improve strength-related paper properties by decreasing the presence of lipophilic extractives in the pulp and avoiding pitch as a result [8]. According to some authors [11], the advantages of biopulping and enzyme treatments more than offset the increased investments and running costs involved.

Enzyme treatments have been applied to various raw materials prior to their conventional delignification [12]. Thus, eucalyptus wood was treated enzymatically prior to kraft pulping, Berrocal et al. [13] applied enzymes to wheat straw before soda pulping, and Akhtar et al. [14] used them to produce thermomechanical birch pulp. Introducing enzymatic treatment in the biomass biorefining process is expected to facilitate the selective separation of polysaccharides in the raw materials [7,15,16,17,18].

Obviously, using an enzymatic treatment to disrupt lignocellulosic materials can also have some disadvantages. Cellulose is a highly crystalline polymer, whereas lignin is especially recalcitrant to dissolution and hinders penetration of enzymes in the material [19,20,21]. In order to circumvent this shortcoming, in this work we used an autohydrolysis treatment to facilitate the selective subsequent extraction of most of the hemicellulose fraction of the material [3,4,22,23,24] and the penetration of the enzyme, with a view to increasing the delignification efficiency.

The pulp and paper sector has traditionally questioned the usefulness of autohydrolysis treatments on the grounds that adequate strength-related properties in paper can only be obtained from pulp containing a certain amount of hemicelluloses. However, conventional delignification processes remove a sizeable proportion of hemicelluloses in the black liquor [25]. In fact, the most labile fraction of the hemicellulose–lignin complex is rapidly lost during kraft pulping: 15–25% of all lignin and 40% of all hemicellulose, according to Núñez [26] and Villar [27].

In this work, we used a laccase/mediator system following the autohydrolysis of eucalyptus wood prior to its soda–anthraquinone pulping and examined the impact of the pretreatment on the properties of the resulting pulp. The enzymatic and autohydrolysis conditions were chosen in such a way as to maximize hemicellulose extraction while preserving glucan integrity. The properties of pulp made from the solid phase remaining after autohydrolysis and enzymatic treatment of *Eucalyptus globulus* wood were compared with those of conventionally produced soda–anthraquinone pulp.

## 2. Materials and Methods

### 2.1. Characterization of Raw Material

In this work, we used *Eucalyptus globulus* wood from a plantation in Huelva, Spain that was ground and sieved through an 8 mm mesh (a particle size known to result in no diffusional constraints). The ground, sieved material was homogenized and air-dried, and appropriate samples were stored for analysis. Following further grinding to a particle size of less than 0.5 mm, the samples were analysed for moisture (Tappi T264-cm-07 [28] and Tappi T204 cm-07 [29]), hot water soluble (Tappi T207 cm-08 [30]), 1% NaOH soluble (Tappi T212 om-02 [31]), and acid-soluble lignin (Tappi UM 250 [32]). Additionally, they were subjected to quantitative acid hydrolysis with 72% H_2_SO_4_ (Tappi T249-cm-09 [33]). The wet weight after storage was 12–25%. The hydrolysis treatment provided a solid residue corresponding to klason lignin and a liquid supernatant containing acetic acid and sugars (glucose, xylose, and arabinose) that were determined by high-performance liquid chromatography (HPLC). This chromatographic determination was performed using an Agilent 1100 HPLC, (Agilent Technologies Hewlett-Packard-Strasse, Waldbronn, Germany) equipped with an ion-exchange resin BioRad Aminex HPX-87H column under the following conditions: mobile phase, 0.005 mol·L^−1^ of sulphuric acid; flow rate, 0.6 mL·min^−1^; and column temperature, 50 °C. The volume injected was 20 µL

From the data of these concentrations, taking into account stoichiometric corrections and the decomposition of sugars, the content in polymers (glucan, xylan, araban, and acetic acid) that were hydrolyzed to give the monomers measured in the analyzed sample is calculated. Hemicellulose fraction were determined by difference between holocellulose and α-cellulose fractions.

### 2.2. Autohydrolysis Process: Pulping Procedure and Formation of Paper Sheets after Enzymatic Treatment

The autohydrolysis treatment was conducted under the optimum conditions established in previous works [34,35], namely, a liquid/solid ratio of 8 kg water/kg raw material, a temperature of 180 °C, and an operating time of 30 min (in these works, autohydrolysis temperature was 181–250 °C and operating time was 30–60 min; a liq/solid ration between 6/1 and 8/1 was used in isothermal and non-isothermal conditions). 

These operating conditions can be expressed according to the severity factor (*R*_0_) introduced by Overend and Chornet [36]. The *R*_0_-factor is given by the following equation: (1)$$R_{0}=t\;\exp^{\left(\frac{T-100}{14.75}\right)}$$
where residence time (*t*) is in minutes and the reaction temperature (*T*) is given in °C. The numerical constants in the expression are related to the activation energy for removal of xylan during process. In our case, *R*_0_ = 6776.

The treatment was performed in a 10 L stainless steel reactor from MK Systems, Inc (North Adams, MA, USA). Reaching the operating temperature in the reactor took 43 min. The reactor allowed for the recycling of its contents and external heating or cooling of the fluid. In this way, it ensured uniform mixing of eucalyptus chips with the cooking fluid. Once the autohydrolysis treatment was finished, the reaction mixture was cooled to 25 °C before the reactor was opened.

Solid fraction from autohydrolysis process was subsequently subjected to an enzymatic treatment according to the experimental design enzymatic-mediator treatment. In this solid fraction yield, glucan, hemicellulose, klason lignin, and acid-soluble lignin contents were determined according the same procedures used for raw material.

The lacassa (Trametres versicolor) activity was assessed by measurement of enzymic oxidation of 2,20-azinobis-(3-ethylbenzothiazoline-6-sulphonic acid) (ABTS) at 436 nm (ε436 = 29.300 M^−1^·cm^−1^) in 0.1 M sodium acetate buffer at pH 5 at room temperature. One unit of enzyme activity is defined as the amount of enzyme that oxidizes 1 mmol ABTS in 1 min [37].

Subsequently, a process of soda-anthraquinone pulping was performed and the corresponding sheets of paper were produced. Pulps were obtained in the same reactor used for the hydrothermal pretreatment.

According to a previous work [35], the following operational conditions were used in the pulping process: operation temperature: 143 °C; operation time: 85 min; soda concentration: 13% (expressed as % NaOH as dry weight); liquid/solid rate: 8/1; and kg water/kg raw material and anthraquinone concentration: 0.1% (dry weight).

Following cooking, the pulp was separated from liquor using a grid tray with 0.16 mm mesh and then disintegrated without breaking the fibers, for 3 min. Pulping yield, kappa number, and viscosity were determinate according to Tappi standard procedures Tappi T257 sp-14 [38], Tappi T236 cm-85 [39], and Tappi T230 om-99 [40], respectively. Paper sheets were prepared with an ENJO-F-39.71 sheet machine according to the Tappi T205 sp-95 [41] standard. From paper sheets, tensile index (Tappi T494 om-01 [42]), burst index (Tappi T403 om-10 [43]), tear index (Tappi T414 om-04 [44]), and brightness (Tappi T525 om-12 [45]) were determined. The rate of heating is the same that was described in autohydrolysis process.

### 2.3. Experimental Design Enzymatic-Mediator Treatment: Multiple Regression Model

Surface response analysis in combination with a suitable experimental design is a well known methodology for statistical modelization and optimization. The statistical basis of this methodology can be found in classic works [46,47]. For example, the work of Jiménez et al. [48] shows a more detailed application of the methodology that is not repeated here. In our case, 3 independent variables (laccase concentration, mediator concentration (in this study, 1-hydroxybenzotriazole (HBT) was used as mediator), and time of process) were used to analyze the linear and quadratic influence on the dependent variables. Additionallky, the interaction terms were considered (yield, glucan, hemicellulose, klason lignin, and acid-soluble lignin contents). In this work, an enzymatic process was modeled. To carry out this modelling, with the minimum testing, a 2*n* central composite experimental design was used (*n* is the number of independent variables (3 in our case). The total number needed experiments is 16 with a central point replicated). With experimental results, a second-order polynomial in the independent variables can be obtained. Previously, normalization of independent variables allows for an easy comparison of coefficients of the independent variables. The normalized points of experimental design are shown in Table 2. For normalization of variables, the following equation was used:(2)$$X_{k}^{N}=\frac{X_{k}-\bar{X}}{\left(X_{max}-X_{min}\right)/2}$$
where $X_{k}^{N}$ is normalized value (−1, 0 and +1) of variable *X*_k_ (for example, values of 60, 120, and 180 for time). $\bar{X}$ is the average value of the variable, and *X_max_* and *X_min_* are its maximum and minimum values, respectively.

According to previous experiences with various operational ranges (data and results not shown), the selected operational conditions (3 variables at 3 levels) are shown in Table 1.

The number of tests required was calculated as *N* = 2*^n^* + 2·*n* + nc, with 2*^n^* being the number of points constituting the factor design, 2·*n* that of axial points, and nc that of central points. Under our conditions, *N* = 16.

The experimental results were fitted to the follog second-order polynomial:(3)$$Y=a_{0}+\overset{n}{\underset{i=1}{\sum }}b_{i}\cdot X_{ki}^{N}+\overset{n}{\underset{i=1}{\sum }}c_{i}\cdot X_{ki}^{N}{}^{2}+\overset{n}{\underset{i=1;j=1}{\sum }}d_{i}\cdot X_{ki}^{N}\cdot X_{kj}^{N}(i<j)$$
where *Y* are the dependent variables; $X_{k}^{N}$ are the normalized independent variable; and *a*_0_, *b_i_*, *c_i_*, and *d_i_* are constant unknown characteristics, estimated from experimental data.

The independent variables used in the equations relating to both types of variables were those having a statistically significant coefficient (viz. those not exceeding a significance level of 0.05 in the Student’s *t*-test and having a 95% confidence interval excluding zero). The results were assessed with STATISTICA 10.0 (StatSoft, Inc., Tulsa, OK, USA). As global model adjustment statisticians, squared-R and Snedecor-F have been used. Levels of squared r higher than 0.85 or Snedecor-F higher than 5 were suitable. Figure 1 shows a scheme of all experimental work.

## 3. Results and Discussion

Our main working hypothesis was that pre-hydrolysis + enzymatic treatment of the raw material allows for handsheets with strength-related properties similar to those of conventional commercial paper to be obtained and a hemicellulose fraction extracted for additional exploitation, all under milder industrial operating conditions (viz., lower chemical loads and temperatures, and shorter processing times). Paper sheets were not prepared after enzimatic stage was carried out because the main effect of the enzyme is to facilitate posterior delignification.

The authors have already explored this hypothesis in previous works, where only autohydrolysis was studied as pretreatment [34]. That pretreatment happened under optimal conditions using a liquid/solid ratio of 8 kg water/kg raw material, a temperature of 180 °C, and an operating time of 30 min. Then, these operating conditions for the autohydrolysis of *Eucalyptus globulus* wood were chosen in this work for the first step of biorefining scheme. In these operating conditions, high solid yield was achieved (approximately 84%). Additionally, it was observed that the glucan initially present in the raw material remained unchanged to a large degree (>80%) and that the polysaccharides of the hemicellulosic fraction of the original raw material were basically solubilized (the xylan, araban, and acetyl groups). Between 65% and 82% of the hemicelluloses present in the raw material was recovered in the liquid phase [49].

On the other hand, the chemical characterization of raw material was carried out in previous works. Briefly, in this work from Loaiza et al. [34], for the *Eucalyptus globulus* wood used, its major fraction was glucan, which accounted for 42.8%; this was followed by klason lignin, with 21.2%; the acid-soluble lignin content, with 6.8%; and hemicelluloses (calculated as the combination of xylan, arabinan, and acetyl groups), with 21.3%. Based on this, *Eucalyptus globulus* could be a suitable raw material for industrial production of hemicellulosic sugars, pulp and paper, and other chemicals.

Table 2 shows the normalized values of the independent variables and summarizes the solid-phase properties obtained by using the proposed experimental designs with enzymatic treatment (laccase/mediator HBT-treatment) after the autohydrolysis process was proposed. Each experimental value was the average of five results for pulp properties. Deviations for the respective means were less than 5%. An overall mass balance of autohydrolysis + laccase treatment can be calculated by multiplying the first column in Table 2 and autohydrolysis yield (0.84).

The results were modeled by using the multiple regression methodology described in experimental design enzymatic-mediator treatment, producing the equations shown in Table 3.

Based on the maximum and minimum values of Table 2 and the contents in raw material, the acid-soluble lignin is more easily attacked by enzymes. The acid-soluble lignin content in the obtained solid phase was 1.8% to 3.5%. Then, the acid-soluble lignin was solubilized between 73.5% and 63.2% with respect to the raw material content. The klason lignin content in the obtained solid phase was 17.3–20.9%. Then, the klason lignin was solubilized between 18.4% and 1.4%. Changes in the other dependent variables were much less substantial. The yields of the enzyme treatment were very high (94–95%), and the yields of the globlal process (autohydrolysis and enzymatic treatments) were 79–80%. This had also been observed in previous studies [50,51]. An increased dissolution of hemicelluloses and lignin could decrease yield, but in fact cellulose losses were reduced in process with previous autohydrolysis, and total yields were scarcely affected.

To better envisage the influence of the operational variables on yield, glucan, hemicelluloses, and lignin contents in solid phase after the treatment with laccase, the response surfaces of Figure 2, Figure 3 and Figure 4 were constructed. The space between the two response surfaces represents the whole range of possible values for each dependent variable; between the two extremes, values of selected independent variables represent both surfaces.

Figure 2 shows the glucan results in the solid phase at two different levels of HBT concentration (+1 and −1). The highest concentrations of glucan in the resulting solid were obtained with medium concentrations of mediator and extreme concentrations of laccase (especially the latter). Additionally, the glucan concentration was independent of the operating time in terms of the constancy of the laccase concentration.

As with glucan, the klason lignin content of the solid (Figure 3) was better with low concentrations of mediator. Additionally, the best responses were those obtained with concentrations on either end of the laccase range; however, the lignin content was slightly lower with *X*_1_ = +1. At a given laccase level, the lowest lignin contents were obtained with extreme values of time but the contents changed very markedly with the operating time.

Acid-soluble lignin (Figure 4) evolved very similarly to klason lignin, with an especially adverse effect of high concentrations of mediator.

The content in hemicellulose of the material that was subjected to the enzymatic treatment (figure not shown) was fairly low, because this component was largely removed by the hydrothermal treatment; also, it changed little with the operating conditions.

As noted earlier, examining the response surfaces allowed for the optimum specific laccase and mediator concentrations (viz. those providing a solid with suitable chemical properties for producing pulp) to be identified. This required maximizing the content in glucan and minimizing that in klason lignin—the hemicellulose content was only 1.6–1.9%, so it did not influence to what extent the solid was exploited.

For optimum results, the mediator concentration should fall on the lower end of the interval, and so should the operating time, in order to save material and energy.

The optimum conditions for delignification with the laccase–mediator combination were thus a mediator concentration of (–1), an operating time of (–1), and a laccase concentration of (+1). Under these conditions, the experimental models of Table 3 estimated a yield of 94.7%, and glucan, klason lignin, acid-soluble lignin, and hemicellulose contents of 36.5%, 18.1%, 1.3%, and 1.6%, respectively.

As stated in the previous sections, we obtained soda–anthraquinone pulp from the solid phase of a hydrothermal treatment of eucalyptus wood using a liquid/solid ratio of 8, a cooking temperature of 180 °C, and a cooking time of 30 min, followed by an enzymatic treatment with laccase (35%) and HTB (1.5%) for 60 min (i.e., the optimum operating conditions established with the experimental design).

The pulping conditions used in the third alkaline delignification step were as follows: 13% NaOH, a temperature of 143 °C and an operating time of 85 min. Table 4 shows the results for the cellulose pulp thus obtained and the paper sheets made from it. 

The strength of our sheets was compared with that of paper obtained by conventional alkaline delignification of the same material with or without a prior autohydrolysis step. The aim was to test the starting hypothesis that autohydrolysis and enzymatic delignification would provide paper of similar quality to that obtained with the conventional process but conducted under milder conditions in the delignification step.

Table 5 shows the operating conditions used and compares the properties of the resulting pulp and paper with those obtained by the authors in other previous work using the same raw material [34,35]. In Table 5, it can be observed than we found similar or better properties of soda-AQ pulps and paper sheets in pulping process with previous autohydrolysis by using lower operational conditions than in the conventional process without autohydrolysis.

The pulping yield obtained with the autohydrolysis and enzymatic treatments was quite high and similar to those for untreated eucalyptus pulp (in the region of 73.7% with [NaOH] = 21%, *T* = 153 °C, and *t* = 115 min) [35]. Thus, high yields were the result of pretreatments affording more efficient usage of the material and reducing the proportion of uncooked residues.

Applying a treatment of autohydrolysis and laccase mediator enables the acquisition of kappa numbers that are considerably lower (32.9) than those coming from non-pretreated eucalyptus (57 or 47 if the wood was previously subjected to autohydrolysis). 

The pre-treatments allow one to reduce the kappa number to 15%, facilitating the subsequent alkaline delignification of the material but using more moderate operating conditions.

Although some authors have found that kraft pulp obtained with a hemicellulose dissolution treatment loses some strength, this is not the case if the pulp is first hydrolysed and then delignified with a soda–anthraquinone mixture, which can be as strong as pulp from untreated material [35].

Tensile strength, other physical properties of pulp, and intrinsic viscosity are related to one another with the degree of polymerization of the polysaccharides. In our case, the intrinsic viscosity follows the same trend as the tensile index. The value reported in Table 4 is similar to those of Loaiza [34] for eucalyptus wood subjected to soda delignification without or with previous autohydrolysis (between 485.6 and 926.1 mL·g^−1^). With a soda paste process with previous treatments of autohydrolysis and laccase-mediator, sheets of paper with a tensile strength of 10.4 N·m·g^−1^ can be obtained. This value is within the range of values (6.5–17.7 N·m·g^−1^) obtained by means of a conventional delignification system with soda-AQ without pretreatments. In general, under the selected process conditions, the strength properties of our pulps are slightly below traditional Eucalyptus Kraft pulps but our values are similar to those of Martín-Sampedro [22] for eucalyptus wood subjected to hydrothermal steaming or steam explosion and subsequent delignification with a conventional kraft procedure (8 to 16 N·m·g^−1^).

The burst index was also seemingly favoured by the two fractionation steps. Thus, the resulting value—1.12 Mpa·m^2^·kg^−1^—is similar than the typical values for conventional processes (0.45–1.12 Mpa·m^2^·kg^−1^) or even those involving an autohydrolysis pretreatment (0.59–1.32 Mpa·m^2^·kg^−1^).

Our tear index—0.57 mN·m²·g^−1^—is similar to that for pulp obtained with or without autohydrolysis (0.55–1.42 and 0.79–1.37 Mpa·m^2^·kg^−1^, respectively).

In summary, the strength-related properties of the paper were not substantially altered by whether the raw material was subjected to autohydrolysis and an enzymatic treatment with laccase and a mediator. Rather, these steps facilitated delignification in a subsequent step and the use of milder operating conditions. This allowed us to obtain soda–AQ paper of similar strength to that of sheets produced from untreated pulp, albeit with a lower soda concentration (13% vs. 21%), and a lower temperature (143 vs. 153 °C) and shorter operating time (85 vs. 115 min). These conditions can reduce costs and provide a liquid fraction rich in hemicellulose sugars for valorization, for example, with furfural production, plastic derivatives of xylose, fermentable media, etc.

## 4. Conclusions

Biorefining lignocellulosic materials enables their integral exploitation by allowing various fractions to be obtained with successive treatments. Thus, using a soda concentration of 13% NaOH, a temperature of 143 °C, and an operating time of 85 min, after autohydrolysis and enzymatic treatments, provides paper sheets with a tensile strength of 10.4 N·m·g^−1^, a burst index of 12 Mpa·m^2^·kg^−1^, and tear index of 0.57 mN·m^2^·g^−1^. These values are similar to those for paper produced by conventional soda pulping but can be obtained with a 38% lower soda concentration, a 6.5% lower temperature, and a 26% shorter operating time.

Additionally, the prehydrolysis and enzymatic delignification of the raw material prior to the delignification with soda-AQ results in paper sheets with lower kappa number and brightness and strength properties similar to conventional soda-AQ paper and a liquid fraction rich in hemicellulose compounds that can be used in additional ways. The advantage of this biorefinery scheme is that it requires a lower concentration of chemical reagents, and shorter operating times and temperatures in the alkaline delignification stage, which represents an economic and environmental improvement over the conventional process.

## Figures and Tables

**Figure 1 polymers-11-00731-f001:**
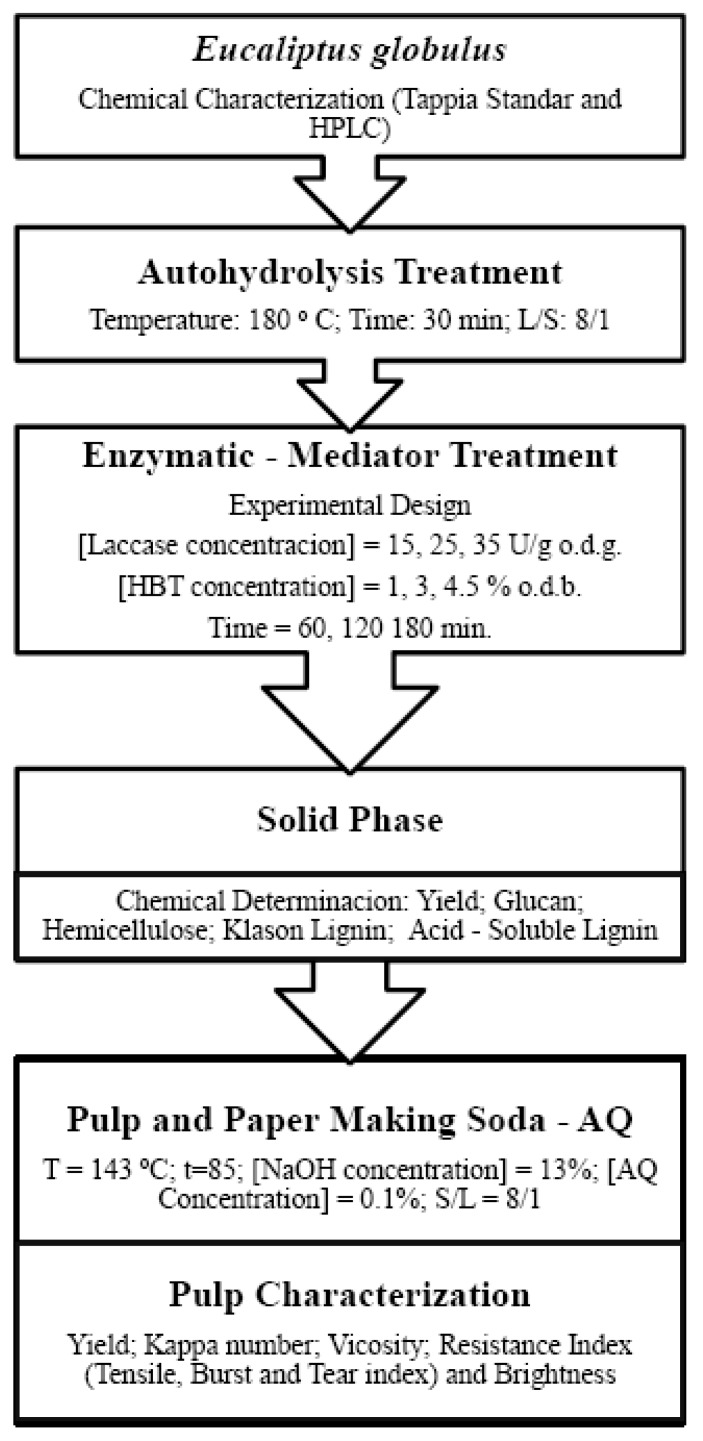
Scheme of experimental work.

**Figure 2 polymers-11-00731-f002:**
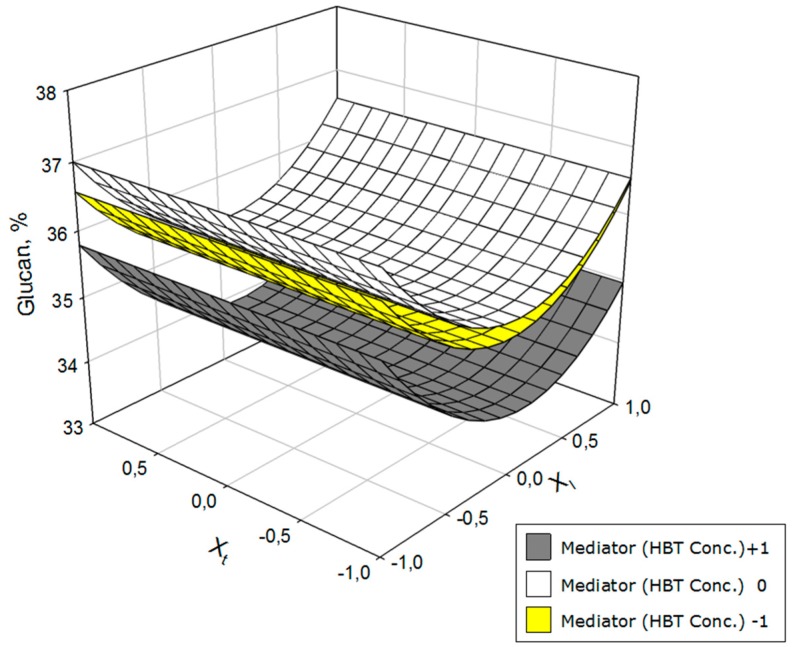
Variation of the glucan as a function of laccase concentration (*X*_l_) and operation time (*X*_t_) at three HBT concentration levels.

**Figure 3 polymers-11-00731-f003:**
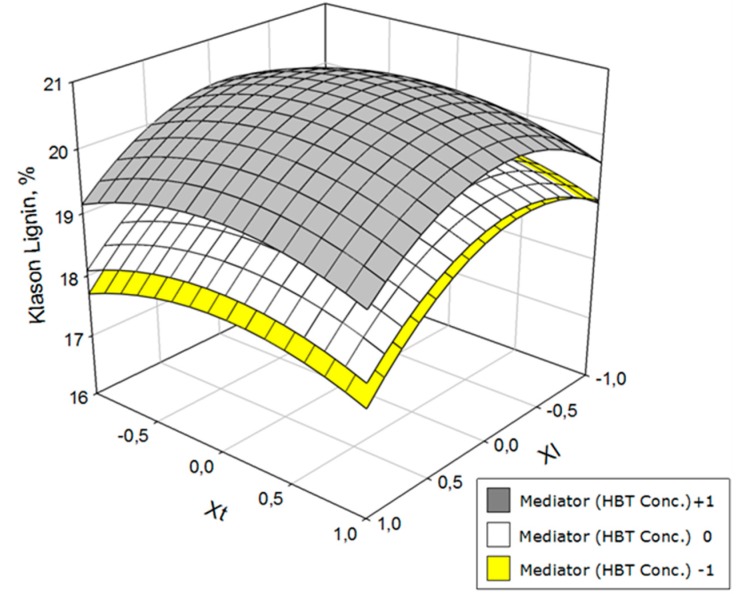
Variation of the klason lignin as a function of laccase concentration (*X*_l_) and operation time (*X*_t_) at three HBT concentration levels.

**Figure 4 polymers-11-00731-f004:**
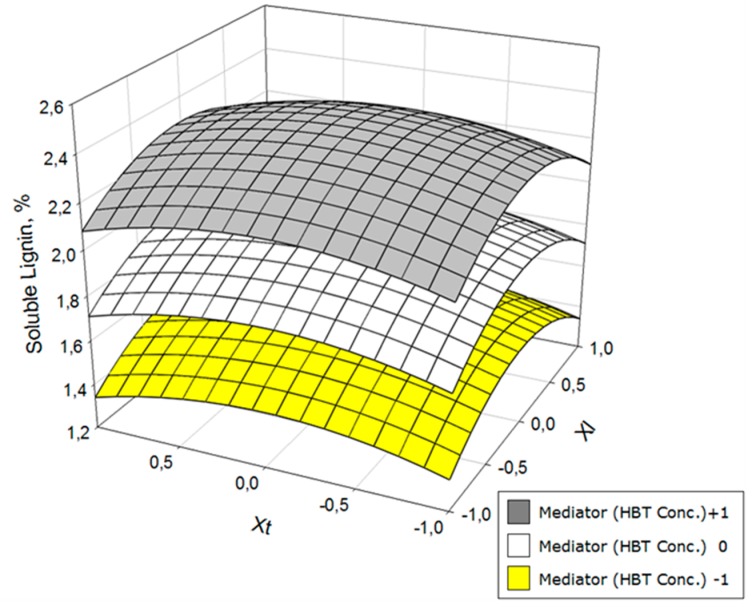
Variation of the acid-soluble lignin as a function of laccase concentration (*X*_l_) and operation time (*X*_t_) at three HBT concentration levels.

**Table 1 polymers-11-00731-t001:** Operating conditions and variables of experimental design enzymatic-mediator treatment.

Conditions	−1	0	1
Laccase concentration (*X*_l_), U/g o.d.b.	15	25	35
HBT concentration (*X*_m_), % o.d.b.	1.5	3	4.5
Time (*X*_T_), min.	60	120	180
Liquid/solid relation	8		
Temperature, °C	45		
pH	4.5		
Impregnation time, min.	30		

**Table 2 polymers-11-00731-t002:** Results for autohydrolysis + enzymatic yield, glucan, hemicelluloses, and lignin contents in solid phase after the treatment with laccase. Normalized values of independent variables (*X*_l_: laccase concentration; *X*_m_: 1-hydroxybenzotriazole (HBT) concentration, and *X*_t_: operation time).

Normalized Values			Responses				
Laccase Concentration	HBT Concentration	Operation Time	Enzymatic Yield %	Glucan %	Hemicelluloses %	Klason Lignin %	Soluble Lignin %
0	0	0	94.4	35.5	1.9	20.2	2.8
0	0	0	94.5	35.4	1.9	19.8	2.9
1	1	1	94.6	35.1	1.7	18.7	3.1
1	1	−1	94.8	34.8	1.8	19.4	2.9
1	−1	1	94.7	36.6	1.8	18.2	1.8
1	−1	−1	94.1	36.4	1.7	17.3	1.8
−1	1	1	94.5	35.9	1.6	19.5	3.0
−1	1	−1	94.8	35.8	1.6	19.7	3.1
−1	−1	1	94.5	36.2	1.7	19.3	2.1
−1	−1	−1	94.7	37.0	1.6	18.8	2.0
1	0	0	94.4	36.5	1.8	18.7	2.7
−1	0	0	94.4	36.7	1.7	19.3	2.7
0	1	0	94.3	33.7	1.9	20.9	3.5
0	−1	0	94.1	34.3	1.8	19.5	2.4
0	0	1	94.9	35.5	1.9	19.5	2.8
0	0	−1	95.0	35.3	1.9	19.3	2.8

**Table 3 polymers-11-00731-t003:** Equations obtained for each dependent variable of autohydrolysis + enzymatic process after autohydrolysis process. Dependent variables: Y_yld_: enymatic yield (%); YGlu: glucan (%); YHem: hemicelluloses (%); Ylk: klason lignin (%); Y_ls_: acid-soluble lignin (%). Independent variables: *X*_l_: laccase concentration; *X*_m_: HBT concentration; and *X*_t_: operation time.

Equation	Adjusted R²	F- Snedecor
Y_yld_ = 94.44 − 0.02 *X*_l_ + 0.07 *X*_m_*X*_m_ + 0.44 *X*_t_*X*_t_ + 0.11 *X*_l_*X*_t_ − 0.11 *X*_m_*X*_t_	0.81	11.6
YGlu = 35.32 − 0.23 *X*_l_ – 0.58 *X*_m_ + 1.45 X_l_ *X*_l_ − 0.82 *X*_m_*X*_m_ − 0.19 *X*_l_*X*_m_	0.94	31.8
YHem = 1.89 − 0.06 *X*_l_ – 0.15 *X*_l_*X*_l_ − 0.05 *X*_m_ *X*_m_ + 0.02 *X*_l_*X*_m_ − 0.04 *X*_m_*X*_t_	0.97	82.0
Y_lk_ = 19.91 − 0.42 *X*_l_ + 0.51 *X*_m_ − 0.93 *X*_l_*X*_l_ + 0.34 X_m_X_m_ − 0.48 *X*_t_*X*_t_ + 0.21 *X*_l_ *X*_m_	0.95	15.1
Y_ls_ = 2.03 + 0.37 *X*_m_ − 0.21 *X*_l_*X*_l_ − 0.11 *X*_t_*X*_t_	0.95	94.0

**Table 4 polymers-11-00731-t004:** Chemical characterization of pulp and physical properties of paper sheets obtained from pulp after autohydrolysis and enzymatic treatment.

Chemical Properties of the Pulp	Physical Properties of the Handsheets
Yield = 73.3%	Brightness ISO% = 24.7
Kappa Number = 32.9%	Tensile Index = 10.4 N·m·g^−1^
Intrinsic Viscosity = 751.3 cm³·g^−1^	Burst Index = 1.1 Mpa.m²·kg^−1^
Lignin = 11.5%	Tear Index = 0.6 mN·m·g^−1^
Glucan = 76.1%	-

**Table 5 polymers-11-00731-t005:** Results of soda-AQ pulping with Eucalyptus with and without previous autohydrolysis.

	Eucalyptus without Autohydrolysis	Eucalyptus with Autohydrolysis
Soda-AQ pulping conditions	Temperature: 153–173 °C; time: 65–115 min, NaOH: 13–21%	Temperature: 143–163 °C; time: 40–90 min, NaOH: 9–17%
Yield (% over raw material. Dry basis)	57.1–73.7	42.5–67.2
Kappa n°	32.9–76.7	33.2–87.9
Brightness (%)	11.8–26.8	12.01–31.93
Tensile index (N·m·g^−^^1^)	6.5–17.5	11.31–23.34
Burst index (MPa·m^2^·kg^−1^)	0.45–1.12	0.59–1.32
Tear index (mN·m^2^·g^−1^)	0.55–1.42	0.79–1.66

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
