# Peer review of "Optimization of Laccase/Mediator System (LMS) Stage Applied in Fractionation of Eucalyptus globulus"

_polymers, 2019, doi:10.3390/polym11040731_

Round 1

Reviewer 1 Report

The manuscript shows relevant results for its publication. 

Some aspects should be taken into account:

- Please, include severity factor of autohydrolysis treatment

- Please, include lacasse activity methods.

- Please, include chemical composition of autohydrolysis liquor

Author Response

Polymers

Ref.: Polymers-478977

Title: Optimization of Laccase/mediator system (LMS) stage applied in fractionation of Eucalyptus globulus

The authors thank the reviewer for his comments.

The corrections are the following (list of changes):

Reviewer 1

- Please, include severity factor of autohydrolysis treatment

 Line 87 (now lines 91-96). The next sentence was added: “These operating conditions can be expressed according to the severity factor (R0) introduced by Overend and Chornet [36], The R0-factor is given by the equation:

                                                                                                          (1)  

Where residence time (t) is in minutes and the reaction temperature (T) is given in ºC. The numerical constants in the expression are related to the activation energy for removal of xylan during process. In our case, R0 = 6776.

The reference: Sixta H., 2006 Handbook of Pulp. Wiley-VCH Verlag GmbH & Co. KGaA, Weinheim, Germany. Vol 1: 189-191 was added.

References have been updated

- Please, include lacasse activity methods.

Lines 102-105 (now lines 106-109). The next paragraph was added: “The lacassa (Trametres versicolor) activity was assessed by measurement of enzymic oxidation of 2,20-azinobis-(3-ethylbenzothiazoline- 6-sulphonic acid) (ABTS) at 436 nm (ε436 = 29.300 M_1 cm_1) in 0.1 M sodium acetate buffer at pH 5 at room temperature. One Unit of enzyme activity is defined as the amount of enzyme that oxidizes 1 mmol ABTS in 1 min [37].”.

-The next reference:

Moldes, D., Vidal, T. Laccase–HBT bleaching of eucalyptus kraft pulp: Influence of the operating conditions.  Bioresource Technology 2008, 99, 8565–8570.

Was added. References have been updated

- Please, include chemical composition of autohydrolysis liquor

This article focuses on the evaluation of the enzymatic stage of delignification. Chemical composition of autohydrolysis liquor was not evaluated. It is indicated in section 2.2: "The autohydrolysis treatment was conducted under the optimum conditions established in previous works [34, 35]".

The authors have similar results from other works. Specifically, In these conditions, 85.0 % of glucan, 9.2 % of xylan and 86.5 % of lignin rest in solid phase after autohydrolysis.  Remainder fractions and degradation products were in liquid phase. But authors do that with other wood lot and prefer not include these results in this article.

Reviewer 2 Report

The work has an innovative aspect. The combination of autohydrolysis and enzymatic delignification stages for alkaline pulping of eucalyptus wood. It can be an important advance in biorefineries development. In this sense, the introduction explain in a suitable form this idea and it is seem correct to me.

The most controversial aspect is the need for the presence of hemicelluloses in the pulp. With various fractionation stages, the hemicelluloses are intensively removed in the autohydrolysis and delignification processes and they will not be available to provide resistance to paper sheets.

The authors raise this problem in introduction section. In fact, in my opinion, it is the more important study of the work and the results could be more extensive, but the experimental is correct and conclusions are suitable supported by data.  The authors argue that hemicelluloses content in final pulp could be similar to a conventional process. Then, the article is interesting for a first advance in this stages combination. I think, it’s a suitable article for publication in “Polymers”.

-Line 74. The HPLC analitical method was not provided.

-Line 77. What were the ashes determined for?.

-Lines 80 to 149. A briefly description about optimum conditions calculated in autohydrolysis must be provided.

-Line 127. What previous experiences (data and results not shown)?. It’s confuse.

-Lines 151 to 155, Have the authors thought about obtaining paper directly after the enzymatic stage?

-Table 2. It is not clear if the results from the table are after the treatment with laccase or treatment with autohydrolysis and enzyme (see also table 3).

-Line 189. A “space” is lack.

-Line 191. The parenthesis is not necessary.

-Line 230. ¿Table 4 or table 3?. Results in lines 224-231 are not the same of table 4 below.

Author Response

LIST OF CHANGES

Polymers

Ref.: Polymers-478977

Title: Optimization of Laccase/mediator system (LMS) stage applied in fractionation of Eucalyptus globulus

The authors thank the reviewer for his comments.

The corrections are the following (list of changes):

Reviewer 2

The work has an innovative aspect. The combination of autohydrolysis and enzymatic delignification stages for alkaline pulping of eucalyptus wood. It can be an important advance in biorefineries development. In this sense, the introduction explain in a suitable form this idea and it is seem correct to me.

The most controversial aspect is the need for the presence of hemicelluloses in the pulp. With various fractionation stages, the hemicelluloses are intensively removed in the autohydrolysis and delignification processes and they will not be available to provide resistance to paper sheets.

The authors raise this problem in introduction section. In fact, in my opinion, it is the more important study of the work and the results could be more extensive, but the experimental is correct and conclusions are suitable supported by data.  The authors argue that hemicelluloses content in final pulp could be similar to a conventional process. Then, the article is interesting for a first advance in this stages combination. I think, it’s a suitable article for publication in “Polymers”.

 - Line 74. The HPLC analitical method was not provided.

Line 74 (now lines 77-80). The paragraph: “This Chromatographic determination was performed using an Agilent 1100 HPLC equipped with an ion-exchange resin BioRad Aminex HPX-87H column under the following conditions: mobile phase, 0.005 mol·L-1 of sulphuric acid; flow rate, 0.6 mL·min-1; and column temperature, 50 °C. The volume injected was 20 µL”. was added. (Lines 77-80)”.

- Line 77. What were the ashes determined for?.

Lines 77-79 (now lines 83-84): “Ashes were determined by calcination (Tappi T211 om-12[34]) and hemicellulose fraction by difference between holocellulose and α-cellulose fractions.” was changed by: “Hemicellulose fraction were determined by difference between holocellulose and α-cellulose fractions”.

In addition the reference [34] was deleted. “TAPPI T211-om-12. Ash in wood, pulp and paper, and paperboard: combustion at 525ºC, TAPPI Press, Atlanta, G.A. 2012.”

References have been updated.

-Lines 80 to 149. A briefly description about optimum conditions calculated in autohydrolysis must be provided.

Line 83 (now lines 88-90. The sentence: “(in these works autohydrolysis temperature ranges between 181-250 ºC, operating time of 30-60 min and liq/solid ration between 6/1 and 8/1 were used in isothermal and non-isothermal conditions)” was added.

- Line 127. What previous experiences (data and results not shown)?. It’s confuse.

Line 127 (now line 146): “According to previous experiences...” was changed by: “According to previous experiences with various operational ranges...” .

- Lines 151 to 155, Have the authors thought about obtaining paper directly after the enzymatic stage?

Line 155 (now line 172-173). The next sentece was added. “Paper sheets after enzimatic stage not was carried out because the main effect of the enzyme is facilitate the posterior delignification”.

-Table 2. It is not clear if the results from the table are after the treatment with laccase or treatment with autohydrolysis and enzyme (see also table 3).

Table 2 and table 3: “enzymatic” was changed by “autohydrolysis + enzymatic”.

-Line 189. A “space” is lack.

Line 189 (now line 207). A “space“ was added before “3.5”.

-Line 191. The parenthesis is not necessary.

The parentheses was deleted.

-Line 230. ¿Table 4 or table 3?. Results in lines 224-231 are not the same of table 4 below.

 (*) Line 230 (now line 253): “Table 4” was changed by “Table 3”.

Reviewer 3 Report

The present manuscript reported the use and optimization of a laccase/mediator system treatment after an autohydrolysis process prior to a soda-anthraquinone pulping with the goal of obtaining a biorefinery scheme that requires a lower concentration of chemical reagents, shorter operating times and temperature. The experiments were well designed, and the results were well discussed. This is a good paper but could be improved by addressing a few questions/responding to a few comments.

1- Overall, authors should pay more attention to the punctuation marks, the place of the references, as well as some minor grammar mistakes. Some examples of that are given below:

-        Line 36: According to some authors [11], instead of [11].

-      Line 40: Thus, [12], ... I think you should place this reference in the previous sentence and remove one of the two comas.

-        Line 65: Eucalyptus globulus should be in italics. Please check it in the rest of the text.

-        Line 82: previous work(S) [35,36] --> two references.

-        Line 142: Figure 1 show(S)....

-        Line 161: “high solid yield” instead of “yield high solid”

-       Line 167: Briefly, in this work from Loaiza et al. [35] instead of Briefly, in these works from Loaiza et al. [35]. There is just one reference.

2- Lines 40-42: Please rewrite this statement.

3- Line 55: I think the reference 26 should be placed at the end of the sentence together the reference 27.

4- Line 74: Please define HPLC for the first time.

5- Please number the equations.

6- Lines 109-110: Please rewrite this statement.

7- Lines 120-121: Please remove this sentence: “Table 2 shows the normalizes points of experimental design.” You say it later before the appearance of the Table 2, which is better.

8- Lines 145-149: I think authors should remove this paragraph from the experimental section and place it in the results and discussion section before the figures 2, 3 and 4.

9- Line 230: I think authors refers to Table 2 instead of Table 4.

10- Line 232: Table 4 should be place within the section 3.1. Properties of the resulting paper sheets” and not before.

11- Please, if possible, improve the quality of the Figure 1.

12-Acid-soluble lignin instead of just soluble lignin (Table 2) because after you refers to acid-soluble lignin in the text.

13- Fix inappropriate capitalization throughout paper.

14- The authors say that this biorefinery scheme, among other things, can provide a liquid fraction rich in hemicellulose sugars for its valorization. Thus, authors should add some examples of that using similar processes in order to improve the last part of the discussion.

Author Response

LIST OF CHANGES

Polymers

Ref.: Polymers-478977

Title: Optimization of Laccase/mediator system (LMS) stage applied in fractionation of Eucalyptus globulus

The authors thank the reviewer for his comments.

The corrections are the following (list of changes):

Reviewer 3

The present manuscript reported the use and optimization of a laccase/mediator system treatment after an autohydrolysis process prior to a soda-anthraquinone pulping with the goal of obtaining a biorefinery scheme that requires a lower concentration of chemical reagents, shorter operating times and temperature. The experiments were well designed, and the results were well discussed. This is a good paper but could be improved by addressing a few questions/responding to a few comments.

1- Overall, authors should pay more attention to the punctuation marks, the place of the references, as well as some minor grammar mistakes. Some examples of that are given below:

-  Line 36: According to some authors [11], instead of [11].

Line 36: [11]. was changed by [11],

- Line 40: Thus, [12], ... I think you should place this reference in the previous sentence and remove one of the two comas.

Line 40: The sentence “Thus, [12],” was changed by “ [12]. Thus,”

- Line 65: Eucalyptus globulus should be in italics. Please check it in the rest of the text.

Line 65 (now line 67). The sentence “Eucalyptus globulus” was corrected to italic along the document.

- Line 82: previous work(S) [35,36] --> two references.

Line: 82 (now line 87): "work" was changed by "works"

- Line 142: Figure 1 show(S)....

Line: 142 (now line 163): “show” was changed by “shows” (line 149).

- Line 161: “high solid yield” instead of “yield high solid”

Line 161 (now line 179): The sentence “yield high solid” was changed by “high solid yield” (line 164).

-Line 167: Briefly, in this work from Loaiza et al. [35] instead of Briefly, in these works from Loaiza et al. [35]. There is just one reference.

Line 167 (now line 185): “These works” was changed by “this work”.

2- Lines 40-42: Please rewrite this statement.

Lines 40-42: “…treated eucalyptus wood enzymatically prior to kraft pulping [13] applied enzymes to wheat straw before soda pulping and [14], used them to produce thermomechanical birch pulp. Introducing an enzymatic treatment in a biomass biorefining process is expected to facilitate the selective separation of polysaccharides in the raw material [7, 15- 18].“ was changed by: “…treated eucalyptus wood enzymatically prior to kraft pulping, Berrocal et al. [13] applied enzymes to wheat straw before soda pulping and Akhtar et al. [14], used them to produce thermomechanical birch pulp. Introducing an enzymatic treatment in a biomass biorefining process is expected to facilitate the selective separation of polysaccharides in the raw material [7, 15- 18].”

3- Line 55: I think the reference 26 should be placed at the end of the sentence together the reference 27.

Line 55 (now line 56): The sentence “[26] 15–25 % of all lignin and 40% of all hemicellulose according to [27]).” was changed by: “ 15–25 % of all lignin and 40% of all hemicellulose according to Núñez [26] and Villar [27].”

4- Line 74: Please define HPLC for the first time.

Line 74 (now line 76): The sentence “High Performance Liquid Chromatography (HPLC)” was added.

5- Please number the equations.

The enumeration of the equations was added.

6- Lines 109-110: Please rewrite this statement.

Lines 109-110 (now lines 126-128): “Surface response analysis in combination with a suitable experimental design, is a well know methodology for statistical modelization and optimization. The statistical basis of this methodology could be found in classic works [44, 45]. For example, the work of Jiménez et al. (1999) [46] shows a”... was changed by: “Surface response analysis in combination with a suitable experimental design, is a well known methodology for statistical modelization and optimization. The statistical basis of this methodology could be found in classic works [46, 47]. For example, the work of Jiménez et al.  [48] shows a....”.

7- Lines 120-121: Please remove this sentence: “Table 2 shows the normalizes points of experimental design.” You say it later before the appearance of the Table 2, which is better.

Lines 120-121 (now lines 138-139): The sentence “Table 2 shows the normalizes points of experimental design” was changed by “The normalizes points of experimental design it is shown in table 2”.

8- Lines 145-149: I think authors should remove this paragraph from the experimental section and place it in the results and discussion section before the figures 2, 3 and 4.

The next paragraph in lines 145 to 149 was changed to lines 215-219: “To better envisage the influence of the operational variables on yield, glucan, hemicelluloses and lignin contents in solid phase after the treatment with laccase, the response surfaces of Figures 2 to 4 were constructed. The space between the two response surfaces represents the whole range of possible values for each dependent variable between the two extremes values of selected independent variable for represent both surfaces.”

9- Line 230: I think authors refers to Table 2 instead of Table 4.

See comment (*) on reviewer 2

10- Line 232: Table 4 should be place within the section “3.1. Properties of the resulting paper sheets” and not before.

Table 4 was placed after section 3.1 on line 263

11- Please, if possible, improve the quality of the Figure 1.

Quality of figure 1 was improved.

12-Acid-soluble lignin instead of just soluble lignin (Table 2) because after you refers to acid-soluble lignin in the text.

“Soluble lignin” was changed by “acid-soluble lignin” along the text.

13- Fix inappropriate capitalization throughout paper.

Inadequate capitalization throughout the text was corrected.

14- The authors say that this biorefinery scheme, among other things, can provide a liquid fraction rich in hemicellulose sugars for its valorization. Thus, authors should add some examples of that using similar processes in order to improve the last part of the discussion.

Lines 288-289 (now lines 314-316). The sentence: “These conditions can reduce costs and provide a liquid fraction rich in hemicellulose sugars for valorization” was changed by: “These conditions can reduce costs and provide a liquid fraction rich in hemicellulose sugars for valorization, for example with furfural production, plastic derivatives of xylose, fermentable media, etc.”

Also (change from authors), the sentence: “In this work, we used Eucalyptus globulus wood from a plantation in Huelva, Spain, that was ground and sieved through an 8 mm mesh  - a particle size known to result in no diffusional constraints” in lines 67-69, was changed by: “In this work, we used Eucalyptus globulus wood from a plantation in Huelva, Spain, that was ground and sieved through an 8 mm mesh (a particle size known to result in no diffusional constraints)”.
